# Exploiting Suspension Plasma Spraying to Deposit Wear-Resistant Carbide Coatings

**DOI:** 10.3390/ma12152344

**Published:** 2019-07-24

**Authors:** Satyapal Mahade, Karthik Narayan, Sivakumar Govindarajan, Stefan Björklund, Nicholas Curry, Shrikant Joshi

**Affiliations:** 1Department of Engineering Science, University West, 46132 Trollhättan, Sweden; 2International Advanced Research Center for Powder Metallurgy and New Materials, Hyderabad 500069, India; 3Treibacher Industrie AG, 9330 Althofen, Austria

**Keywords:** titanium carbide, chromium carbide, suspension plasma spray, wear

## Abstract

Titanium- and chromium-based carbides are attractive coating materials to impart wear resistance. Suspension plasma spraying (SPS) is a relatively new thermal spray process which has shown a facile ability to use sub-micron and nano-sized feedstock to deposit high-performance coatings. The specific novelty of this work lies in the processing of fine-sized titanium and chromium carbides (TiC and Cr_3_C_2_) in the form of aqueous suspensions to fabricate wear-resistant coatings by SPS. The resulting coatings were characterized by surface morphology, microstructure, phase constitution, and micro-hardness. The abrasive, erosive, and sliding wear performance of the SPS-processed TiC and Cr_3_C_2_ coatings was also evaluated. The results amply demonstrate that SPS is a promising route to manufacture superior wear-resistant carbide-based coatings with minimal in situ oxidation during their processing.

## 1. Introduction:

Wear is a severe problem in a vast majority of industrial applications, leading to reduced durability of engineering components and increased frequency of replacement shutdowns. Protective coatings of carbide-based compositions have been extensively employed to enhance wear resistance, and the popular processing routes to produce them have included physical vapor deposition (PVD), chemical vapor deposition (CVD), thermal spray, etc. [1]. Among these, thermal spraying offers unique advantages such as the ability to spray thicker protective layers, faster deposition rates, and cost efficiency. Historically, atmospheric plasma spraying (APS), high velocity oxy-fuel (HVOF) spraying, and more recently high-velocity air-fuel (HVAF) spraying techniques have progressively become the thermal spray methods of choice for depositing carbide-based coatings [2]. A significant limitation of these techniques is their inability to process fine powders due to considerable feeding-related challenges [3], although employing such feedstock can potentially result in improved tribological performance of the coatings [4]. 

Suspension plasma spraying (SPS) is an advancement in APS, enabling spraying of fine feedstock (100 nm–5 µm diameter) [5], and has already shown promise for yielding thermal barrier coatings (TBCs) with improved performance compared to APS [6]. SPS coatings can similarly enhance wear performance due to unique microstructural features inherently associated with the process, such as the smaller splat size, fine scale porosity, low surface roughness, etc. An illustrative description of distinct microstructures resulting from the thermal spraying of different feedstock sizes, as well as their likely influence on wear performance, is shown in Figure 1. Yet, the predominant focus of SPS research so far has been on TBCs. In particular, there have been very few prior efforts to deposit pure carbide coatings by SPS, which are ideal candidates for mitigating wear. Recently, Mubarok et al. reported SPS-processed SiC coatings, which exhibited excellent carbide phase retention post spraying [7]. Berghaus et al. reported similar findings (minimal oxidation) for SPS-processed WC-Co coatings, demonstrating the capability of SPS to minimize the in-flight oxidation of carbide feedstock [8]. The reason for the minimal oxidation of carbides in SPS-processed coatings compared to APS can be attributed to the presence of a solvent (ethanol or water) which consumes a considerable part of the plasma energy during its evaporation, thus minimizing feedstock decomposition.

Recognizing the above, the present study specifically deals with titanium carbide (TiC) and chromium carbide (Cr_3_C_2_) coatings (without any metal or alloy binder) produced by SPS. To the best of our knowledge, no prior attempts have been made to deposit TiC- and Cr_3_C_2_-based coatings via the SPS route. The SPS carbide coatings were comprehensively characterized using SEM, XRD, porosity, hardness, etc. The tribological behavior of the coatings under erosive, sliding, and abrasive wear modes was also evaluated.

## 2. Experimental Work

SSAB Domex® 350LA (low-alloyed steel, SSAB AB, Stockholm, Sweden) substrates were grit blasted to provide a surface roughness of approximately 3 µm Ra prior to spraying. Two experimental water-based suspensions, comprising 40 wt.% TiC and Cr_3_C_2_, respectively, were produced by the Treibacher Industrie AG (Althofen, Austria) for this work. Water-based suspensions were chosen due to their higher surface tension versus ethanol, which promotes the formation of relatively denser coatings [9]. Particle size analysis of the liquid feedstock was performed using CILAS 1064 equipment (CILAS, Orleans, France). An Axial III plasma torch (Mettech Corp., Vancouver, Canada) was employed to deposit the coatings. The spray parameters used for SPS deposition of both the coatings are given in Table 1. A gas mixture comprising argon, nitrogen, and hydrogen was used as the primary gas.

The surface morphology and cross-sectional microstructure of the coatings were analyzed using a scanning electron microscope (HITACHI TM-3000, Tokyo, Japan) in back-scattered electron (BSE) mode. Surface roughness was examined using a stylus-based profilometer (MITUTOYO SURFTEST-301, Kawasaki, Tokyo, Japan). Micro-hardness measurements were performed on the cross section of coatings (HMV-2 series, SHIMADZU Corp., Tokyo, Japan). Hardness testing was performed using a load of 980.7 mN (HV_0.1_) and the dwell time was kept at 15 s. Ten independent measurements were made, and their mean and standard deviation values are reported herein. Porosity measurements were made using ImageJ software (version 1.52p, University of Wisconsin, Wisconsin, US)) [10] by considering fifteen different cross-sectional SEM micrographs at 5000× magnification. XRD analysis of the top surface of as-sprayed coatings was performed using a high-energy intensity micro X-ray diffractometer (RAPID-II-D/MAX, Rigaku Corp., Tokyo, Japan). A slow scan rate comprising a step size of 0.01° and time of 10 s per step was utilized. 

The as-sprayed specimens were polished to a surface roughness <1 µm on the Ra scale prior to wear testing. Three different wear tests (erosion, sliding, and abrasion) were performed in this work. The erosion test (TR-470, DUCOM, Bengaluru, India) was performed at room temperature according to the ASTM G76-13 standard [11] at an impingement angle of 90°. Alumina of approximately 50 ± 10 µm mean particle size was used as the erodent media. During the test, the air pressure was maintained at 0.5 bar and the specimens (1-inch diameter coupons) were exposed to the erosion test for 10 min. The abrasive-wear test was performed as per ASTM G65 standard [12] using a dry abrasion test rig from DUCOM (DUCOM instruments, Bangalore, India, 2019) with 80 mesh silica sand particles at a load of 5 kg. The speed of the wheel was kept at 245 ± 5 rpm and the abrasive flow rate was kept at 350 g/min. For the sliding-wear tests, specimens of size 10 mm ×10 mm × 4 mm were cut from the coated plates. Sliding wear tests were performed as per ASTM G99 standard [13] by employing a coated pin (sintered WC-6Co) of diameter 6 mm, at a load of 5 kgf sliding against a disc rotated at 5 m/s velocity. The duration of this test was three hours and twenty minutes. In each of the above wear tests, the weight loss was measured using a high-accuracy weighing balance (Sartorius, Cubis® II, Sartorius Gmbh, Göttingen, Germany), accuracy: 0.01 mg). Three samples were used for erosion tests, while only one specimen was used for sliding- and abrasive-wear tests.

## 3. Results and Discussion

The particle size distribution of titanium carbide and chromium carbide feedstock is shown in Figure 2a,b respectively. The median (D_50_) particle size of titanium carbide was approximately 2.21 µm, and the D_90_ was approximately 4.30 µm. The median (D_50_) particle size of chromium carbide was 3.84 µm and D_90_ was 6.25 µm. The particle size of chromium carbide was only slightly larger than that of titanium carbide, which is not expected to significantly influence splat size in the deposited coatings.

The SEM micrographs of TiC and Cr_3_C_2_ powders revealed that they were both comprised of irregularly shaped particles (see Figure 3a,b). The Cr_3_C_2_ particles were clearly seen to be relatively coarser than the TiC particles, confirming the accompanying particle size distribution results, according to Figure 3b.

The top surface views of SPS-processed TiC and Cr_3_C_2_ coatings in Figure 4 show fine-structured splats in the size range 3–4 µm, as compared to conventional powder-derived splats, which are at least an order of magnitude larger. The reason for the finer splat size in SPS can be directly attributed mainly to the considerably smaller particles in the feedstock. Moreover, the relatively lower droplet momentum in the case of SPS ensures that the droplets flatten to a lesser extent on impact with the substrate compared to conventional spray processes such as APS. This results in a much-refined microstructure in the case of the suspension-derived SPS coatings in comparison to the powder-derived APS coatings. The top-view SEM micrograph of the TiC coating shows unmolten spherical particles (see Figure 4a). The Cr_3_C_2_ coating also showed fewer unmolten particles than those observed in TiC coating (Figure 4b). The reason for lower unmolten particles in Cr_3_C_2_ (melting point: ~1800 °C [14]) than TiC could be attributed to the significantly lower melting point of Cr_3_C_2_ compared to TiC (melting point: ~3100 °C). Furthermore, in both the SPS-processed coatings, the top surface morphology did not resemble the cauliflower-like microstructure as reported elsewhere for SPS-processed YSZ coatings, which accompanies the columnar microstructures desired for TBC applications [6].

The low-magnification cross-sectional SEM micrograph of an SPS-processed TiC coating is shown in Figure 5a, and reveals a largely homogeneous microstructure. The corresponding high-magnification cross-sectional SEM micrograph is more revealing, and shows uniformly distributed porosity with very few unmelted particles (Figure 5b). Furthermore, the splat boundaries between successive splats could be discerned, but no inter-pass porosity was obvious, suggesting reasonably good inter-splat cohesion. Additionally, the cross-sectional SEM micrographs did not show any delamination cracks or separation at the splat boundaries, indicating good coating integrity.

The cross-sectional SEM micrograph of the Cr_3_C_2_ coating at low-magnification in Figure 6a also showed uniform distribution of porosity. The high-magnification cross-sectional SEM micrograph in Figure 6b shows inter-splat boundaries between successive splats. At certain locations, the splat boundaries in the Cr_3_C_2_ coating were observed to reveal separation between splats, indicating relatively poor cohesion. Notwithstanding the very promising results exhibited by the Cr_3_C_2_ coating as discussed subsequently, the above microstructural examination suggests that there clearly exists room for further optimization of coating quality by modifying suspension formulation and/or manipulating spray parameters. 

Phase analysis of both transition metal carbide coatings by micro-XRD revealed varying extents of oxidation and decarburization occurring in-flight, to show different levels of sub-carbides and oxides in Figure 7a,b. Among various carbide materials, TiC is known for its difficulty in melting and is quite stable over a wide compositional range, between TiC_0.97_ and TiC_0.50_. It is highly challenging to retain the carbides during plasma spraying [15,16]. However, in the present case, the strongest peaks corresponded to titanium oxy-carbide and TiC phases (Figure 7a). This implies minimal decarburization, although considerable amounts of titanium oxides were noted to have formed during the SPS deposition process. The predominant presence of carbide phases can potentially provide better wear resistance by reducing the friction coefficient between the sliding contacts, which was confirmed from the sliding wear results (Table 2). Furthermore, the XRD pattern of the SPS TiC coating in Figure 7a suggests that there could be some amorphous phase formation. Li et al. have reported similar findings related to amorphous phase content in tungsten-carbide-based coatings deposited by HVOF [17]. 

Similarly, plasma/HVOF spraying of Cr_3_C_2_ typically results in phase transformation into Cr_7_C_3_ and Cr_23_C_6_. It has also been previously reported that Cr_7_C_3_ forms as a rim over Cr_3_C_2_ particles [15]. On the other hand, the completely molten chromium carbide (melting point: 1811 °C [14]) preferably precipitates in the form of stable M_23_C_6_-type carbides of Cr_23_C_6_, which are relatively smaller in grain size [18]. Accordingly, Cr_7_C_3_ and Cr_23_C_6_ phases were detected in case of SPS Cr_3_C_2_ coatings along with non-stoichiometric CrC phases and oxides of chromium, as shown in Figure 7b.

The TiC coating (12.9 ± 2.2 µm) showed a higher surface roughness (Ra) in as-sprayed condition than the Cr_3_C_2_ coating (4.6 ± 1.2 µm). It should be mentioned that the TiC feedstock had a relatively lower median particle size (D_50_) and lower splat size in the as-sprayed condition than did Cr_3_C_2_. Furthermore, both the coatings were deposited using identical spray parameters. One possible explanation for the observed difference in Ra values could be the difference in the degree of in-flight melting of powder particles because of their vastly different melting temperatures. TiC (melting point: 3100 °C) has a higher melting temperature than Cr_3_C_2_ (melting point: 1800 °C), which could result in a higher degree of melting in the case of Cr_3_C_2_ than TiC. As discussed previously, some unmolten TiC particles were also noted in the coating, which could have contributed to higher surface roughness, suggesting need for further feedstock and process optimization. However, it is also pertinent to mention that the surface roughness of the SPS-processed Cr_3_C_2_ coating in this work was lower than that previously reported for HVOF-processed Cr_3_C_2_-based coating [19]. For wear application, it is desirable to produce coatings with low surface roughness. Therefore, SPS seems to be a promising processing method to achieve coatings with low surface roughness due to its finer splats compared to APS, HVOF, etc.

The porosity content of both SPS processed carbide coatings was comparable and measured to be in the range of 6–9%, which is higher than ideally desired for wear applications. However, this preliminary attempt at depositing carbide coatings via SPS constitutes a useful basis to minimize porosity content and further improve coating quality through process optimization. The micro-hardness values for SPS-processed Cr_3_C_2_ (920 ± 70 HV_0.1_) and TiC (980 ± 60 HV_0.1_) coatings were found to be lower than the bulk hardness values for TiC (2850 HV_0.1_) [20] and Cr_3_C_2_ (1834 HV_0.1_) specimens [21,22]. This could be attributed to the features such as pores and inter-splat boundaries that are typical of thermally sprayed coatings. Regardless of the above, the wear performance of these coatings was suggestive of considerable promise, as discussed below.

In the wear tests performed under identical test conditions, SPS-processed Cr_3_C_2_ showed superior erosive and sliding wear resistance to the SPS-processed TiC coating (Figure 8a,c and Table 2). However, the abrasion test results showed comparable performance for the investigated coatings, according to Figure 8b. Note that the hardness for the TiC coating was higher than that of the Cr_3_C_2_ coating. Furthermore, the cross-sectional SEM micrograph of Cr_3_C_2_ showed few regions of poor cohesion between the splats. However, its erosion and sliding wear performance was superior to that of TiC. On the other hand, such fine-structured splats in the SPS-processed carbide coatings (in case of both Cr_3_C_2_ and TiC) could favor improved performance under all wear modes compared to a microstructure with relatively larger splats (APS) for an identical composition. Similar findings related to improved wear resistance compared to conventional coatings was reported by Liang et al. in the case of nanostructured coatings, and the reason was attributed to their enhanced mechanical properties (higher hardness, higher cohesive strength, higher toughness, etc.) [23]. The above results from our preliminary study with carbide suspensions show considerable promise and clearly motivate further, more detailed, investigations. Process optimization and the investigation of different suspension properties (e.g., carbide particle size, solid loading, etc.) and their influence on wear behavior are logical next steps. Furthermore, examining the worn surfaces of the coatings and worn debris (for sliding wear) using SEM/EDS would provide further insights into the wear mechanisms responsible for material removal under different wear modes. These will be performed as a continuation of the present study.

## 4. Conclusions

In this work, we demonstrated for the first time that TiC and Cr_3_C_2_ coatings could be successfully deposited by suspension plasma spray (SPS) utilizing a feedstock of fine powders. The top-view microstructures of both coatings showed splats that were an order of magnitude lower than those typically observed in APS coatings, indicating the possibility of producing coatings with refined microstructures for wear applications. XRD analysis revealed the presence of the desired carbide phases in the deposited coatings, demonstrating the potential of SPS as a facile route to the deposition of carbide-based coatings. The coatings were shown to have good integrity, and the tribological performance of these coatings was found to be extremely promising under different wear modes (i.e., erosion, sliding contact, and abrasion), motivating their further investigation with and without the addition of binder materials.

## Figures and Tables

**Figure 1 materials-12-02344-f001:**
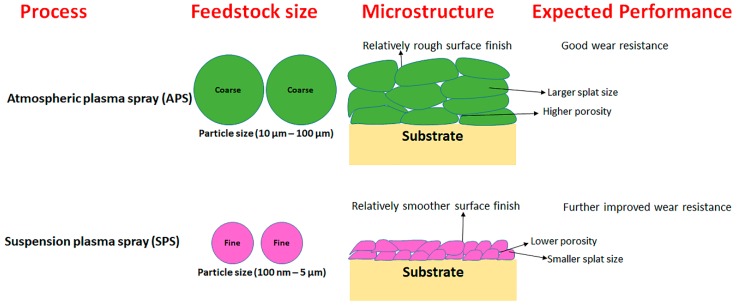
Schematic of comparison of feedstock–microstructure–property relationships in APS and SPS coatings.

**Figure 2 materials-12-02344-f002:**
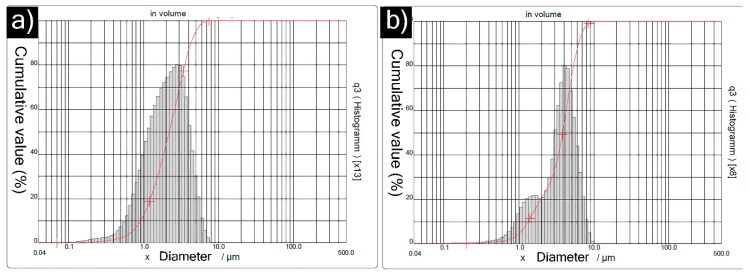
Particle size distribution of (**a**) TiC powder feedstock (**b**) Cr_3_C_2_ powder feedstock.

**Figure 3 materials-12-02344-f003:**
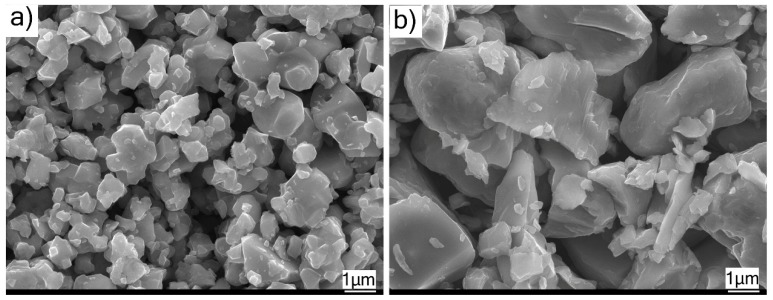
SEM micrographs showing§; (**a**) feedstock TiC powder; (**b**) feedstock Cr_3_C_2_ powder.

**Figure 4 materials-12-02344-f004:**
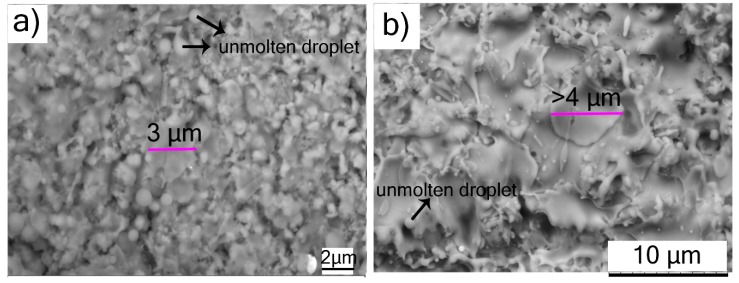
Surface morphology SEM micrographs of SPS-deposited coatings: (**a**) TiC; (**b**) Cr_3_C_2_.

**Figure 5 materials-12-02344-f005:**
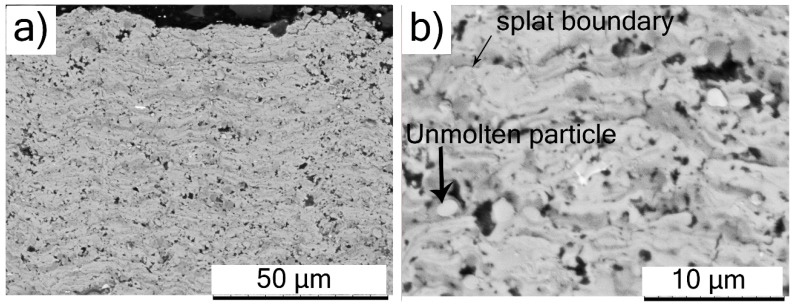
SEM micrographs of cross sections of SPS-deposited TiC coating: (**a**) low magnification; (**b**) high magnification.

**Figure 6 materials-12-02344-f006:**
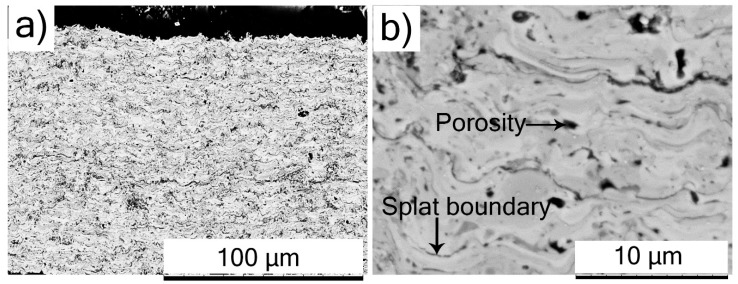
SEM micrographs of cross sections of SPS-deposited Cr_3_C_2_ coating: (**a**) low magnification; (**b**) high magnification.

**Figure 7 materials-12-02344-f007:**
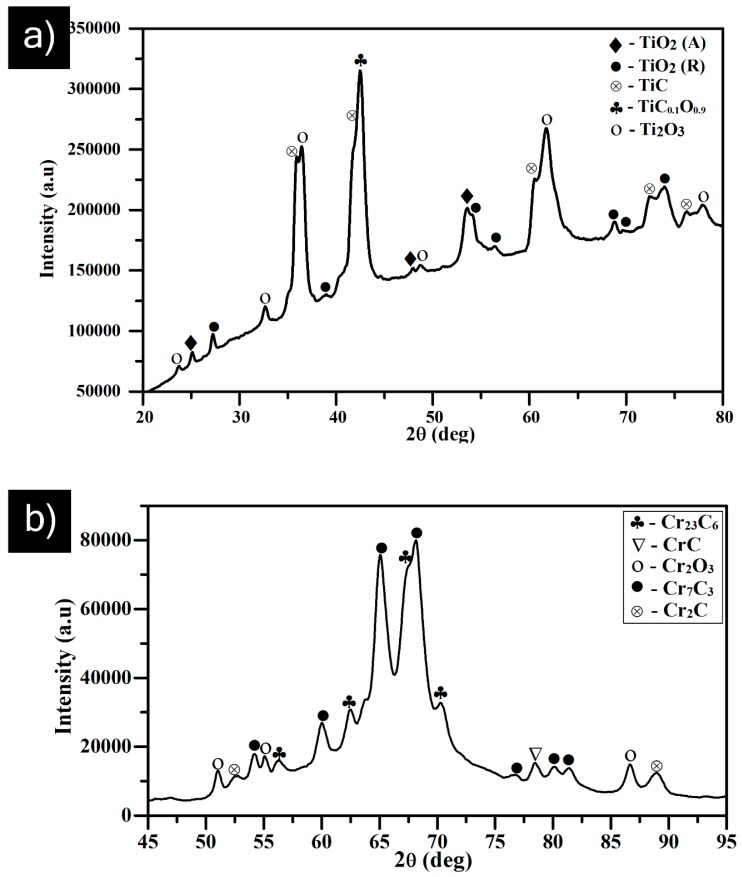
XRD analysis of the as-sprayed surface of (**a**) TiC coating; (**b**) chromium carbide coating.

**Figure 8 materials-12-02344-f008:**
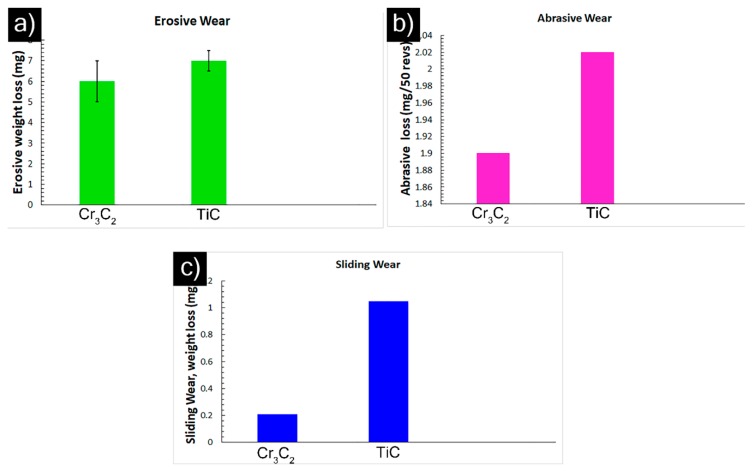
Performance of SPS-processed Cr_3_C_2_ and TiC coatings under different wear modes: (**a**) erosion; (**b**) abrasion; and (**c**) sliding.

**Table 1 materials-12-02344-t001:** Spray parameters used for depositing TiC and Cr_3_C_2_ coatings.

Suspension Feed Rate (mL/min)	Nozzle Diameter (inches)	Spray Distance (mm)	Surface Speed (mm/s)	Atomizing Gas Flow Rate (L/min)	Power (kW)	Enthalpy (kJ/I)	Current (A)
40	3/8	100	100	15	111	9.7	200

**Table 2 materials-12-02344-t002:** Sliding wear results of the investigated coatings.

Sample Identity	Volumetric Wear Loss (mm^3^)	Coefficient of Friction (µ)
Cr_3_C_2_	0.0314	0.60
TiC	0.2129	0.28

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
