# Peer review of "Exploiting Suspension Plasma Spraying to Deposit Wear-Resistant Carbide Coatings"

_materials, 2019, doi:10.3390/ma12152344_

Round 1
Reviewer 1 Report
In this manuscript, the authors tried to demonstrate that suspension plasma spraying (SPS) is a promising process to manufacture good wear resistant carbide coatings. They deposited carbide coatings by using the suspension plasma spraying technology under the same spraying parameters with TiC and Cr3C2 aqueous suspensions, respectively. The resulting coatings were characterized in terms of surface morphology, microstructure, phase constitution and micro-hardness. The abrasive, erosive and sliding wear performance of the SPS processed TiC and Cr3C2 coatings was examined and compared with air plasma sprayed T-400 (Tribaloy) coatings.
The authors emphasized that the novelty of this work relates to processing fine sized carbides in the form of suspensions for the first time to fabricate wear resistant coatings. However, this is not the case. Many research jobs carried out on wear resistant coatings by suspension Plasma Spraying in the past two decades. For instance, the nanostructured WC-Co cermet Coatings by suspension Plasma Spraying, etc.
Titanium and chromium-based carbides are attractive coatings to impart wear resistance. However, plasma spraying of carbide material is a challenging task since the carbides tend to decompose at high temperatures, as shown in XRD results of this work. The loss of carbon is thought to occur because of direct oxidation on the surface of solid carbide, leading to formation of other phases.
How to maintain low carbon loss becomes a key issue. Suspension plasma spraying permits projection of liquid feedstocks directly. The interaction between plasma and liquid atomizes the suspension into a fine mist and evaporates the suspension medium, thereby concentrating the solid content into fine particles. Despite the high plasma temperature, the decomposition of the carbide material should in principle be reduced by the evaporating liquid carrier, which imposes a substantial thermal load on the plasma. Upon impact on the substrate, these particles form thinner lamellae than in conventional spraying. Thin coatings with more refined microstructures and potentially smoother surface finishes are created. If the authors could add some work in the aspects mentioned above, it would be innovative.
More details as shown below
In the manuscript, the abrasive, erosive and sliding wear performance of the SPS processed TiC and Cr3C2 coatings was examined and compared with air plasma sprayed T-400 (Tribaloy) coatings. Since the carbide coatings and T-400 coatings are different in terms of chemical composition and mechanical properties, and deposited by different methods, a comparison sounds not so convincible.
In the abstract, page 1, It is acceptable if the abbreviation is common in the field, there is no risk of confusion. since SPS is explained in the context, it is better to explain APS as well.
In the experimental work, the APS parameters are not presented.
If the plasma operating conditions like current, voltage, primary gas, etc, are listed in table 1, it would be better.
More details are required for XRD measurement in the experimental work.
More details for surface roughness in the experimental work. How many measurements were taken? The result of surface roughness of the as-sprayed carbide coating was not given and discussed in the result and discussion section. (it was only used to measure surface roughness for specimens in the wear test?)
In the experimental work, microhardness test needs more details, like load, dwell time, etc. How many measurements?
How many samples for each group in the wear tests (erosion, sliding and abrasion), in the experimental work?
In figure 4, Page 4, it looks there exist many un-melted particles in the TiC coating.
Why not to use the same magnification for size comparison in figure 4.
The porosity can be measured by image analysis. This will be more reliable to compare the two carbide coatings.
Need to label the unmelted particles and the splat boundaries in figure 5 and 6
In lines 112-114, page 4, why it is suggestive of reasonably good inter -splat cohesion? Because the splat boundaries could be discerned?
In line 134, you cannot guess the relative amounts of phases based upon the relative intensities of the diffraction peaks. sometimes one pattern is more intense because it diffracts X-rays more efficiently.
In figure 7, page 6, an amorphous material may be presented in the mixture. Has the background noise been removed?
In line 142-143, page 6, It is not meaningful to compare different materials processed by different methods.
In figure 8, page 7, the error bars are not given in figure 8b and 8c. Could the error evaluation be performed? Is the experiment repeatable?
APS T-400 is not a good reference to compare the wear behaviour for SPS carbide coatings
In lines 152-154, it is hard to conclude that the microstructure comprising fine structured splats (SPS) favours improved performance under all wear modes compared to the microstructure with relatively larger splats (APS), since the comparison is not reliable.
In the conclusion, the authors did not well summarize the content.
Reviewer 2 Report
The paper describes the effect of suspension coatings. While the results are interesting, there are some modifications needed.
1) The abstract is not properly written. Please re-write the abstract. Some of them should move to the introduction section.
2) Please re-write the conclusion.
3) Please describe the substrate.
Reviewer 3 Report
Comments and Suggestions for Authors
This study mainly focuses on evaluating the wear resistance of the titanium carbide and chromium carbide coatings prepared by suspension plasma spraying (SPS). The topic of this study is interesting; however, the introduction of the manuscript is lack of literature review regarding to the SPS deposited carbide coating, such as silicon carbide and tungsten carbide coatings. The authors should provide more background information and indicate the novelty of the current study in the text. Furthermore, the characterization on the wear track and the wear mechanism is not proposed in the current manuscript, which in my opinion should be the most important part of this study. In addition, I would recommend the manuscript minor checked by a native English speaker to make it more readable. The manuscript could be considered for publication only after major revision. Some of my comments are listed below:
Line 6:
The email addresses of all authors are missing except the corresponding author.
Line 17: “The abrasive, erosive and sliding wear performance of the SPS processed TiC and Cr3C2 coatings was evaluated and compared with APS processed coatings of T-400 (Tribaloy),...”
The full term of “APS” should be added in the text.
1. Introduction
Line 51, Figure 1:
Please revise the word “porosity” in the schematic microstructure to “higher porosity” for the APS and “lower porosity” for the SPS so the readers could clearly see the differences between them. Furthermore, I suggest changing the figure caption to “Schematic of feedstock-microstructure-property relationship of the APS and SPS.”
2. Experimental work
Line 53: “Mild steel substrates were grit blasted to provide a surface roughness of approximately 3 μm Ra prior to spraying.”
Which type of mild steel was used in the current study? Please provide more information in the text.
Line 64, Table 1:
In the table caption, the number “3” and “2” of the “Cr3C2” should be subscript.
Line 65: “The surface morphology and cross sectional microstructure of the coatings were analyzed using an SEM (HITACHI 3000, Japan) in back scattered electron (BSE) mode.”
The full term of “SEM” should be added in the text. In addition, please check whether the model of the SEM is correct (S-3000 or TM-3000?).
Line 72: “Erosion test (TR-470, DUCOM) was performed at room temperature according to ASTM G76-13 standard at an impingement angle of 90 degrees.”
More detail information should be provided regarding to the parameters of the erosion test. For example, what is the material of the solid particle? What is the size of the specimen? What is the air pressure during the test?
Line 74: “Abrasive wear test was performed as per ASTM G65 standard using 80 mesh Silica sand particles at a load of 5 Kg.”
Again, the author should describe the testing parameter in the text as detail as possible.
Line 77: “Sliding wear test was performed as per ASTM G99 standard by employing a coated pin (sintered WC-6Co) of diameter 6 mm, tested at a load of 5 kgf against a disc rotated at 5 m/s velocity.”
What is the duration of the sliding wear test? Please provide more detail if possible.
Line 79: “Erosion resistance was tested against silica erodent (80 mesh size) impacting the coated surface at a velocity of 108 m/s for 1-minute duration.”
Shouldn’t this sentence belong to the above paragraph describing the erosion test?
Line 80: “In each of the above wear tests, the weight loss was measured using a high accuracy weighing balance (Sartorius, Germany, accuracy: 0.01 mg).”
The model of the weighing balance is missing.
3. Results and Discussion
Line 86: “The particle size of chromium carbide was slightly larger than that of titanium carbide, which can influence the splat size in the deposited coatings.”
What could be the reason that causing the size differences between the two particles?
Line 89, Figure 2:
Please change the labels of the X- and Y-axis from German to English.
Line 106, Figure 4:
The size of the splat indicated in the figure does not match with the scale bar of the image, please revise.
Line 111: “The high magnification cross sectional SEM micrograph showed uniformly distributed porosity and very few unmelted particles, according to Fig. 5(b).”
Please indicate the locations of the “unmelted particles” in Fig. 5(b) using arrows.
Line 112: “Furthermore, the splat boundaries between successive splats could be discerned but no inter-pass porosity was obvious, being suggestive of reasonably good inter-splat cohesion.”
Please carefully check the grammar of this sentence. In addition, please indicate the location of “splat boundaries” in the figure.
Line 116: “High magnification cross sectional SEM micrograph showed inter splat boundaries between successive splats, according to Fig. 6(b).”
Please provide more description on the microstructure of the Cr3C2 coating. Furthermore, the author should put more effort describing the microstructural deference between TiC and Cr3C2 coating.
Line 128: “Such phases can potentially provide better wear resistance by reducing the friction coefficient between the sliding contacts, which was confirmed from the sliding wear results, see Table. II.”
Which phases? Is it Titanium oxycarbide and TiC or just TiC? Please be more specific.
Line 131: “On the other hand, the completely molten carbide (melting point : 1811oC [11]) preferably precipitates in the form of stable M23C6 type carbides of Cr23C6 which are relatively smaller in grain size [12].”
Do you mean “On the other hand, the completely molten chromium carbide (melting point : 1811oC [11]) preferably precipitates in the form of stable M23C6 type carbides of Cr23C6 which are relatively smaller in grain size [12].”?
Line 139: “Micro-hardness values for SPS processed Cr3C2 (920±70 HV0.1) and TiC (980±60 HV0.1) coatings were found to be lower than the bulk hardness values for TiC and Cr3C2 specimens [13].”
The microhardness of the bulk materials should also be indicated in the text.
Line 147, Table 8: “Performance of SPS processed Cr3C2 and TiC coatings and under different wear modes (a) erosion, (b) abrasion and (c) sliding.”
The table cation should be revised to “Performance of SPS processed Cr3C2 and TiC coatings and APS processed T-400 coating under different wear modes (a) erosion, (b) abrasion and (c) sliding.” In addition, the labels of the X- and Y-axis are too small and should be enlarged.
Line 152: “It seems that the microstructure comprising fine structured splats (SPS) favors improved performance under all wear modes compared to the microstructure with relatively larger splats (APS).”
The statement is mere conjecture. The authors do not provide any experimental results to support this statement. Furthermore, the micrograph of the APS deposited T-400 coatings is totally missing.
Line 161: “Furthermore, examining the worn surface of coating and worn debris (for sliding wear) using SEM/EDS would provide further insights on the wear mechanism of SPS and APS coatings. This will be performed as a continuation of the present study.”
In my opinion, the key point of the current research topic is studying the wear mechanism of the SPS and APS coatings, which does not included in this manuscript.
Round 2
Reviewer 1 Report
I suggest to accept the paper.
Reviewer 3 Report
Good revision. The authors have successfully address the reviewer’s comments. The revised manuscript has meet the requirement for publication.